# Effect of a Short Message Service Intervention on Excessive Gestational Weight Gain in a Low-Income Population: A Randomized Controlled Trial

**DOI:** 10.3390/nu12051428

**Published:** 2020-05-15

**Authors:** Hannah Holmes, Cristina Palacios, YanYan Wu, Jinan Banna

**Affiliations:** 1Department of Human Nutrition, Food and Animal Sciences, College of Tropical Agriculture and Human Resources, University of Hawaii at Manoa, Honolulu, HI 96822, USA; hjholmes@hawaii.edu; 2Department of Dietetics and Nutrition, Stempel College of Public Health and Social Work, Florida International University, Miami, FL 33199, USA; crpalaci@fiu.edu; 3Office of Public Health Studies, University of Hawaii at Manoa, Honolulu, HI 96822, USA; yywu@hawaii.edu

**Keywords:** pregnancy, text message, nutrition intervention, technology, telehealth

## Abstract

Objectives: The objective of this trial was to investigate the effect of educational short message service (SMS), or text messages, on excessive gestational weight gain (GWG) in a low-income, predominantly overweight/obese population. Methods: Participants (*n* = 83) were mostly overweight/obese women recruited at Special Supplemental Nutrition Program for Women, Infants, and Children (WIC) clinics on the island of O’ahu, Hawai’i at 15–20 weeks gestational age. The intervention group received SMS on nutrition and physical activity during pregnancy designed to help them meet Institute of Medicine (IOM) guidelines for GWG and American College of Obstetricians and Gynecologists guidelines for exercise, respectively. The control group received SMS about general health topics during pregnancy, excluding nutrition and physical activity. Both groups received one text message per week for eighteen weeks. GWG was defined as the difference between the last self-reported weight taken before delivery and participants’ self-reported weight before pregnancy. Differences between study groups were examined using t-tests and Chi-square tests. Linear regression models were used to examine association of GWG with study group and other factors. Results: GWG was similar (*p* = 0.58) in the control group (14.1 ± 11.4 kg) and the intervention group (15.5 ± 11.6 kg). The percentage of participants exceeding IOM guidelines for GWG was similar (*p* = 0.51) in the control group (50.0%, *n* = 17) and the intervention group (60.5%, *n* = 23). Conclusions: GWG was not significantly different between intervention and control groups. Trials that begin earlier in pregnancy or before pregnancy with longer intervention durations and varying message frequency as well as personalized or interactive messages may be needed to produce significant improvements.

## 1. Introduction

Excessive gestational weight gain (GWG) in overweight or obese women is associated with adverse pregnancy outcomes [1]. These outcomes include gestational diabetes mellitus, birth complications, postpartum weight retention, and childhood overweight or obesity [1,2]. About 60% of overweight or obese women in the US experience excessive GWG [3]. Healthy eating and physical activity are important modalities for managing GWG [4,5]. Previous interventions for GWG have yielded inconsistent findings, with interventions failing to improve GWG across all weight groups or at all [6]. The systematic review by Skouteris et al. found that interventions focusing on either nutrition or physical activity were less successful in reducing GWG than those that combined the two approaches [6]. Focus groups of low-income women reveal a lack of knowledge on these topics as barriers for maintaining a healthy weight throughout pregnancy [7].

Low-income women in the US are more likely to be overweight or obese and to enter pregnancy in this condition [8]. Women of low-income groups are also more likely to have greater and excessive GWG [9]. In addition to higher likelihoods of overweight or obesity and excessive GWG, low-income populations are hard to reach with traditional, face-to-face interventions [10]. These programs may not be available in certain geographic areas, or the individual may not be close enough to participate. Even when available, these programs may place burdens on low-income individuals due to transportation, cost, and time requirements.

Much of the literature evaluates face-to-face interventions, which do not address the barriers mentioned above. Technology may be a solution for the barriers and the lack of results in interventions to control GWG. Clinically significant weight loss has been achieved at a lower cost using technology-based interventions [10]. One study revealed that overweight and obese women reported positive experiences using mobile and technology-based tools for health during pregnancy [11]. In another study conducted through the Special Supplemental Nutrition Program for Women, Infants, and Children (WIC), a greater proportion of participants using a weight loss website returned to pre-pregnancy weight than those receiving only standard WIC care [12]. Of the technology-based methods suitable for mobile health interventions, the short message service (SMS), or text messages, is a valuable approach for low-income populations. SMS is low cost to both the sender and the recipient, carries a low participant burden, and allows for easy participant response. Mobile phone ownership is widespread, with about 96% of adults in the United States using or owning a cellular phone [13]. A systematic review on the efficacy of SMS for maternal/infant health showed that this approach can be successful when established theories of behavior change are used and when message content is aligned well with outcomes [14]. There are few studies investigating the effectiveness of SMS messages in nutrition interventions during pregnancy. The objective of this study was to assess the effectiveness of an 18-week SMS intervention promoting nutrition and physical activity delivered to a low-income population of predominantly overweight/obese women in Hawai’i on reducing excessive GWG.

## 2. Materials and Methods

### 2.1. Study Design

This was a parallel, randomized, controlled trial in WIC clinics using educational text messages to prevent excessive GWG in a low-income, predominantly overweight/obese population. The trial is registered on clinicaltrials.gov (NCT04330976).

### 2.2. Setting and Participants

The study took place in four WIC clinics across the island of O’ahu, Hawai’i. Eligible women participating in WIC were made aware of the study and, if interested, screened for eligibility. Eligibility criteria were: (1) 10–20 weeks gestational age and 18 years of age or older at time of recruitment; (2) body mass index (BMI) of 20–45 kg/m^2^ in the first trimester; and (3) possession of a cellular phone with the ability to receive text messages without a charge. Exclusion criteria included: (1) conditions requiring a special diet; (2) multiparous pregnancies; (3) unable to consent to participate; (4) unwilling to be randomized.

A sample size of 80 participants (40 per group) achieves 80% power at a 0.05 significance level to detect the effect size of 0.65 in the difference of continuous outcome variables between two experimental arms. Research assistants enrolled participants, and equal numbers of participants were randomized to the control arm or to the intervention arm using random block sizes (2, 4, or 6) with 26 total blocks. A list of randomization numbers and corresponding IDs was computer-generated. Research assistants were provided with the allocation scheme at the start of the recruitment period and referred to this at the time participants eligible to be enrolled were identified. Participants and WIC staff were not aware of the allocation scheme at the time of selection and enrollment. Participants were allocated an ID sequentially as they were recruited, and this ID was matched with the randomized group. The trial ended after all participants with follow-up had received the full 18-week intervention/control message program and had given birth. Only WIC staff were blinded to the treatment assignments.

### 2.3. Development/Delivery of Intervention

Eighteen messages were developed in line with WIC recommendations for pregnant women and were written at a fifth-grade reading level, as determined using the Flesch-Kincaid formula to ensure readability. Messages were reviewed by a pediatrician with experience in working with low-income groups in Hawai’i. Cognitive testing was conducted via interviews with pregnant participants of WIC (*n* = 5). Cognitive interviews focused on determining whether message text was appropriate for the target population to further ensure cultural relevance. Participants were asked to state the meaning of messages in their own words to improve clarity. The method of cognitive testing has been previously described [15].

Intervention messages were developed based on social cognitive theory and focused on energy intake and physical activity for healthy weight gain during pregnancy. Social cognitive theory explains the reciprocal interactions between people, environments, and behaviors [16]. The main constructs of social cognitive theory are self-efficacy, knowledge, goals, expected results, and perceived facilitators and impediments to behavior [17]. The nutrition messages were designed to promote adequate diet quality and quantity to meet the Institute of Medicine guidelines for weight gain during pregnancy [18]. The physical activity messages focused on recommendations from the American College of Obstetricians and Gynecologists, which encourage women to engage in moderate-intensity activity for 20–30 min per day [5]. Figure 1 shows a theory of change diagram for the SMS messages in promoting lifestyle behavior change. Messages for the control group focused on general health during pregnancy, with topics such as the importance of visiting a physician regularly and achieving adequate sleep.

Examples of Intervention SMS

Nutrition and energy intake messages

“Make half your plate fruits and vegetables. Choose a variety, like spinach, carrots, tomatoes, beans, and peas”.“Eating healthy foods is more important now than ever! You need more protein and iron from meat and beans, and calcium and folic acid from vegetables”.“‘Eating for two’ doesn’t mean eating twice as much. You only need about 300 calories more during the last 6 months of pregnancy”.“Omega-3 fats in seafood are important for you and your unborn child. Salmon, sardines, and trout are high in omega-3 fats”.

Physical activity messages

“To walk more: park far from where you are going, take the stairs instead of the elevator, take your pet for a walk, or talk on the phone while walking”.“Include 2 ½ h each week of physical activity such as walking fast, dancing, gardening, or swimming”.“Tips to move more: dance while you cook, get up in a waiting room and walk up and down the aisles”.

Delivery of messages was automated through the SMS platform EZ Texting [19]. Both groups received one message per week for eighteen weeks. Timing of messages varied. The researchers were able to view whether messages were successfully sent to participants.

### 2.4. Measures/Outcomes

The main outcome of the study was GWG, which was calculated by subtracting the participant’s self-reported weight before pregnancy from the last weight taken before birth. Pre-pregnancy weight and last weight before birth were reported by the participant via questionnaires at the first and the second study visits, respectively. The Institute of Medicine provides healthy GWG guidelines for normal, overweight, and obese BMIs (Table 1).

### 2.5. Statistical Plan

Summary statistics were used to describe the sample characteristics by study group, and *t*-tests and Chi-square tests were performed to examine differences between groups or to determine if Institute of Medicine (IOM) guidelines were met. Linear regression models were utilized to determine if GWG was associated with study group, demographics, and other covariates. Variables that were statistically significant at the 5% level in the bivariate analysis were examined in the multivariable linear models. The multivariate model adjusted for height, age group, weight before pregnancy, and number of children. Since the intervention was randomly assigned, the covariates in the multivariable model were included as precision variables to include statistical precision instead of confounders for the intervention effect. Stepwise regression and backward elimination methods were applied to find factors that were associated with GWG while adjusting for participant’s height. Statistical software R was used for the analysis (version 3.5.1, R Foundation for Statistical Computing, Vienna, Austria).

## 3. Results

Recruitment began in October 2017, and the last follow-up took place in October 2018. After screening for eligibility, participants (*n* = 83) were randomized into intervention (*n* = 42) and control (*n* = 41) groups. By follow-up at the last weight taken before pregnancy, 11 participants had been lost, four from the intervention group (*n* = 38) and seven from the control group (*n* = 34). Reasons for loss to follow-up included moving out of state, miscarriage, and discontinuation of intervention.

At baseline, participants were 27.7 ± 5.3 years old on average, 65.5% were Native Hawaiian, Pacific Islander or American Indian, 54.8% had some college education or more, and 37.8% were employed. Almost all participants used prenatal vitamins (97.6%). The average weight before pregnancy was higher in the intervention group than that in the control group, but this difference was not significant. Other baseline characteristics were comparable between groups (Table 2).

There were no differences in attrition rates between groups, with 17.1% of control participants and 9.5% of intervention group participants not completing the study (*p* = 0.490). There were no significant differences between those who completed the study and those who did not. Mean GWG was similar (*p* = 0.580) in the control group (14.1 ± 11.4 kg) and the intervention group (15.5 ± 11.6 kg). Of all 72 participants with pre-pregnancy weight and follow-up data, 55.6% (*n* = 40) exceeded the IOM guidelines for gestational weight gain. In the control group, 50.0% (*n* = 17) of participants exceeded the guidelines, while 60.5% (*n* = 23) of the intervention group exceeded the guidelines. The difference in number of participants exceeding guidelines between the two groups was not significant (*p* = 0.509). Of normal weight, overweight, and obese women, 77.8%, 51.9%, and 52.8% exceeded the guidelines, respectively. The difference across these groups was not statistically significant (*p* = 0.357) (Table 3).

The multivariable linear regression model showed women aged 35 years or older gained an average of 11.5 kg (95% CI: 1.4, 21.2, *p* = 0.021) more than those aged 18–24 and an average of 12.8 kg (95% CI: 3.9, 21.6, *p* = 0.005) more than those aged 25–34. Older age groups were highly correlated with a greater number of previous children (*p* < 0.001). There were no other significant differences between age groups. Women who already had 3–5 children gained more weight than those who had no children (−7.03 kg; 95% CI: −15.0, 1, *p* = 0.086) or 1–2 children (−6.7 kg; 95% CI: −13.5, 0.05, *p* = 0.051). A greater weight before pregnancy was negatively associated with weight gain during pregnancy; 4.5 kg greater pre-pregnancy weight corresponded with 0.91 kg less of GWG (*p* = 0.016). These results are summarized in Table 4.

## 4. Discussion

In this eighteen-week, randomized, controlled trial, GWG did not differ between the intervention SMS and the control SMS groups. Differences in the number of participants exceeding GWG guidelines were not significant across treatment groups or BMI categories.

Clinical trials have shown that technology-based interventions are able to produce levels of weight loss comparable to face-to-face interventions, with the additional benefit of a lower cost per unit of weight lost [10]. A review and meta-analysis of weight management programs including a text messaging component found that these programs were successful in promoting weight loss [20]. Preconception weight loss has been investigated for the purpose of improving fertility, with interventions incorporating exercise and reduced caloric intake associated with improved markers of fertility [21,22]. Preconception weight loss, whether due to lifestyle changes or medical weight-loss treatment (e.g., bariatric surgery), is shown to reduce the risk of some pregnancy complications, such as pre-eclampsia and gestational diabetes [23]. One study in obese women found that interpregnancy weight loss beyond returning to pre-pregnancy weight was associated with a decreased risk of gestational diabetes in a second pregnancy [24]. Preconception weight loss may be beneficial for decreasing risk of preterm birth, however, the impacts of preconception weight loss on other birth outcomes and offspring health have yet to be fully elucidated [25,26]. Depending on preconception diet quality, preconception weight loss may affect mothers’ nutritional status and, therefore, birth outcomes [27].

### 4.1. Strengths

First, this was a randomized, controlled trial that investigated a novel intervention technique for the problem of excessive GWG in a low-income population. The results of this study add to the small but growing body of literature on technology-based interventions for improving maternal and or infant health. In addition, intervention messages were developed with recognized guidelines, social cognitive theory, and cultural considerations in mind.

### 4.2. Limitations

First, the weight measurements used to determine GWG were self-reported. Multiple studies have found self-reported measures of pre-pregnancy weight to be inaccurate, with participants often underestimating or overestimating their weights. This may result in misclassification of participants into BMI groups [28,29].

Second, the sample size of the study may not have been adequate to detect between-group differences. A more intensive recruitment period across more recruitment locations could allow for enrollment of additional participants. Trials with the ability to accommodate larger sample sizes are needed to gain adequate statistical power to determine differences between control and intervention groups.

Third, this trial included a population of low-income, primarily obese or overweight women in Hawai‘i. The findings of this study may not be generalizable to normal or underweight women or women of varying socioeconomic status. It is worth noting that low-income cellphone owners send/receive more SMS than those of higher incomes, and, therefore, SMS interventions may be more effective and appropriate for this population than a middle- or higher-income population [30].

Fourth, the current study did not measure dietary intake, physical activity, or behavior change over the course of the intervention. Studies that collect information on dietary intake or diet quality may help researchers and clinicians determine problem areas that should be addressed or emphasized in future trials or care. It would also help to determine specifically which food choices were most significantly affected by the corresponding messages (e.g., whether the messages about fish and omega-3s were motivational enough to increase fish consumption). Although the intervention aimed to change dietary and physical activity-related behaviors, the study did not investigate changes in food or physical activity choices.

Lastly, perceptions of weight as they relate to health and readiness for change were not factored into message development or delivery. According to health belief and health promotion models of behavior change, perceptions of the relationships between behaviors and health risks are necessary for health-related behavior change [16,31]. A study of 585 pregnant women found that perceived risk of pre-pregnancy BMI and GWG on maternal and infant health was low in both normal and overweight women [32]. Even if individuals perceive a health-related risk, the transtheoretical model of behavior change explains that individuals are in different stages of changes with regard to particular behaviors [33]. Interventions including components based on these theories, along with the social cognitive theory, may be beneficial for supporting behavior change. The study did not address participants’ perceptions of the health risks of GWG, their stage(s) of behavior change, or willingness to change. Participants may have been in different stages of behavior change, thus some messages may not have been effective for them.

### 4.3. Implications for Future Research

This is one of the first studies of its kind in a low-income, diverse, pregnant population. Future research can build on the current study. One potential method of improving the effectiveness of the intervention is to include additional time periods surrounding and during pregnancy: preconception, early pregnancy, and postpartum. Since excessive GWG is associated with a greater risk of adverse health outcomes in overweight or obese women, interventions beginning before pregnancy may be beneficial for reducing the proportion of women entering pregnancy with overweight or obesity [1]. Trials that encompass more time surrounding pregnancy would also allow for longer interventions, which could help increase postpartum weight loss and lead to fewer women entering a second pregnancy with overweight or obesity. Combining maternal diet and exercise educational materials with information about breastfeeding may improve both infant feeding practices and postpartum weight loss, therefore increasing interpregnancy weight loss. Besides the potential benefits of preconception or postpartum weight loss, earlier interventions would allow time for educational messages about healthy diet and exercise to be repeated and may allow additional time for participants to move into more “ready” states of behavior change.

The current study only sent one text message a week over the eighteen weeks of the trial, totaling eighteen text messages. Participants may benefit from receiving more messages at a frequency higher than once per week. One study investigating the use of text messages in a weight maintenance program found that participants desired to receive at least one text message a day, preferably in the morning to increase daily motivation [34].

This study only gathered participant data at enrollment and follow-up and did not measure engagement with the SMS program. Encouraging responses throughout the intervention may be helpful for engagement and retention. A pilot study investigating an SMS program for managing GWG found that 86% of women in the intervention arm responded to prompts, and the intervention had a non-significant effect on GWG compared to the control [35]. The authors of the same study suggested that encouraging responses to messages may have increased the sense of accountability in participants and provided motivation for reaching goals [35].

The messages included in this trial were tailored to the target population, but not to individual participants. There is evidence that tailoring program messages to individuals can assist in education and encouraging behavior change [36]. One study found that computer-tailored nutrition messages were effective in reducing dietary fat intake [37]. A systematic review of tailored eHealth interventions for weight loss found that tailored interventions increased weight loss modestly in four of six studies considered [38]. Tailoring ranges from including participants’ names in messages to using participant data collected to provide feedback and adjust materials to better suit participants during the intervention. This process may also improve participant engagement and retention on the basis that participants may find that information presented this way is more suited to their needs and will continue to use the program [38].

Combining mHealth or eHealth intervention modalities may also be helpful for effecting changes in behavior. In a pilot study aiming to improve postpartum weight loss, the intervention arm included SMS and a social media peer group and was found to increase weight lost compared to the control group receiving standard care [39]. A cluster randomized controlled trial (RCT) found that an intervention utilizing SMS and a weight loss website was effective in increasing weight lost and proportion of participants returning to pre-pregnancy weight than a control [12].

Since this research is aimed at a low-income population, future studies should also include the costs of such programs and measures of cost-effectiveness.

## 5. Conclusions

This trial to prevent excessive GWG utilized a modality for providing educational information that addressed barriers to program participation for low-income women who are especially at risk for overweight, obesity, and excessive GWG. Evidence-based guidelines and social cognitive theory were used to develop targeted messages, which were distributed via SMS. After eighteen weeks of one SMS per week, no significant difference in GWG or exceeding IOM guidelines was found between intervention and control groups. Future studies with more intense, longer interventions and more robust outcomes, including measurement of dietary patterns and behavior change, are needed to determine the effectiveness and the practical potential of SMS for this purpose.

## Figures and Tables

**Figure 1 nutrients-12-01428-f001:**
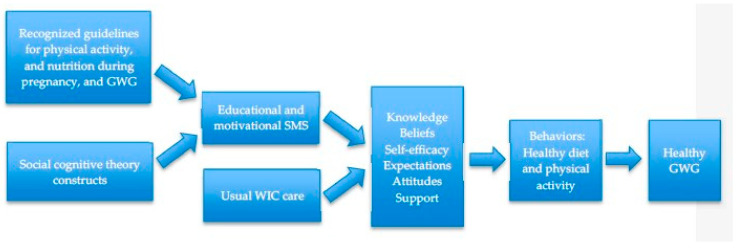
Theory of change diagram for using short message service (SMS) in prevention of excessive gestational weight gain (GWG).

**Table 1 nutrients-12-01428-t001:** Institute of Medicine guidelines for gestational weight gain for singleton pregnancies [17].

	Normal BMI (18.5–24.9 kg/m^2^)	Overweight BMI (25.0–29.9 kg/m^2^)	Obese BMI (Greater Than 30.0 kg/m^2^)
Suggested range (kg)	11.5–16.0	7.0–11.5	5.0–9.0
Excessive (kg)	>16.0	>11.5	>9.0

**Table 2 nutrients-12-01428-t002:** Baseline characteristics of the sample.

	Control (*n* = 41)		Intervention (*n* = 42)		
Characteristics	Mean (*n*)	SD (%)	Mean (*n*)	SD (%)	*p*-Value
Age	27.2	5.51	26.9	5.40	0.748
Number of children	1.27	1.47	1.50	1.33	0.453
Pre-pregnancy weight (kg)	76.2	15.9	80.6	17.7	0.238
Pre-pregnancy BMI	29.8	5.42	30.4	6.04	0.618
Race/Ethnicity ^1^					
Asian	11	26.8	7	16.7	1.00
American Indian	4	9.76	3	7.14	0.392
Black	6	14.6	8	19.1	0.808
Hispanic	8	20.0	17	40.5	0.076
Native Hawaiian	11	26.8	11	26.2	1.00
Pacific Islander	9	22.0	10	23.8	1.00
White	19	46.3	19	45.2	1.00
Education					
Less than college	17	41.5	19	45.2	0.121
Some college	17	41.5	19	45.2	
College or higher	7	17.1	4	9.52	

^1^ Of the 83 participants at baseline, 45 (54.2%) self-identified multiple races/ethnicities.

**Table 3 nutrients-12-01428-t003:** Participants exceeding or not exceeding Institute of Medicine (IOM) GWG guidelines by treatment group and BMI category (*n* = 72).

		Participants Exceeding Guidelines *n* (%)	Participants Not Exceeding Guidelines *n* (%)	*p*-Value
Group	Intervention	23 (60.5%)	15 (39.5%)	0.509
	Control	17 (50.0%)	17 (50.0%)	
BMI	Normal	7 (77.8%)	2 (22.2%)	0.357
	Overweight Obese	14 (51.8%) 19 (52.8%)	13 (48.2%) 17 (47.2%)	

**Table 4 nutrients-12-01428-t004:** Association of GWG with age, weight before pregnancy, height, and number of children.

	Beta (95% CI)	*p*–Value
Intercept	32.5 (18.6, 46.5)	<0.0001
Age 25–34 vs. age 18–24	−1.31 (−7.16, 4.55)	0.662
Age 35+ vs. age 18–24	11.5 (1.7, 21.2)	0.021
Age 35+ vs. age 25–34	12.8 (3.9, 21.6)	0.005
Weight before pregnancy	−0.20 (−0.37, −0.04)	0.016
Height	0.71 (−0.22, 1.65)	0.136
1–2 children vs. none	−0.31 (−6.64, 6.02)	0.924
3–5 children vs. none	−7.02 (−15.03, 0.98)	0.086
3–5 children vs. 1–2 children	−6.6 (−13.5, 0.05)	0.051

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
