# Peer review of "Effect of a Short Message Service Intervention on Excessive Gestational Weight Gain in a Low-Income Population: A Randomized Controlled Trial"

_nutrients, 2020, doi:10.3390/nu12051428_

Round 1

Reviewer 1 Report

Comments:

This is an interesting study of a sms intervention on excessive gestational weight gain in a low-income population. The comments below are restricted to those which reflect suggested changes for improvement.

Abstract
Line 14 add excessive gestational weight gain

Lines 15-17 remove sentence.  Indicates expectation bias.

Line 17 define “WIC”. It may be frequently used initials but so is “SMS” that you choose to define. Be consistent.

Line 20 it is not clear if the SMS that the control group received contained nutrition and physical activity information – how are they different from the intervention? (I know from your methods that they were different, but this should also be clear in the abstract)

Line 22 was the last weight before delivery taken by a health professional? Or was it self-reported like before pregnancy? – When before delivery was it taken? Unclear.

Line 25 move the statistical significance value to after “similar”, i.e. similar (p=0.58) keep the values for each group after the group as you have

Line 27 as above, move the p value after “similar”

Line 29 change “longer intervention” to “longer intervention durations”

Line 29-30 you may want to reconsider this recommendation. Although you recommendation has been previously suggested in the literature, a few studies found that decreased frequency associated with sms personalisation and tailoring and bidirectionality was associated with improved outcomes

Keywords
Line 31 consider the term “telehealth”

Introduction

Line 37 consider specifying country so it is not confusing that you are generalising about global evidence – experience excessive GWG in the US [3].

Line 37 consider changing “proper nutrition” to “healthy eating”

Line 41 insert reference [6] at end of sentence or start the sentence with “The systematic” instead of “A systematic”

Line 44 consider specifying country as in certain non-western countries low-income women may be undernourished

Lines 47-50 you mention geolocation, travel impedance and time as limitations but you do not mention cost. Cost may likely be the greatest limitation for the low-income population. Please include

Lines 66-69 remove. These details belong in the methods section. – replace with the objective, something along the lines of: “The objective of this study was to assess the effectiveness of an 18-week nutrition and physical activity education program, delivered to a low-income, overweight/obese population in Hawai’i via SMS, on the reduction of excessive GWG.”

Methods

Line 86 who performed the randomisation? Research assistants? Statistician? Were they involved in the intervention?

Line 88 report on concealment of allocation

Line 96 “…with experience in working…”

Line 97 what method of cognitive testing was used?

Line 104 “…to meet the Institute…guidelines…”

Line 113 was a variety of colours suggested, when a variety of vegetables was suggested?

Line 119 was caution raised about mercury levels?

Line 135 how was the self-reported weight measured? Did participants specify if they were weighed by a doctor/health care practitioner during e.g. a routine visit? Did they weigh themselves? Did you collect information on the scales used by the practitioner or the participants? Were these scales calibrated? Did the participants wear clothing/shoes? How many layers? Were participants weighed fasted? At a particular time of the day? Had they emptied their bowel and bladder?

Define “last weight taken before birth”. Was this consistently taken for all participants at the same time prior to birth? Was this taken at the clinic/hospital? Was it conducted by a healthcare practitioner? Did you collect information on the scales used by the practitioner or the participants? Were these scales calibrated? Did the participants wear clothing/shoes? How many layers? Were participants weighed fasted? At a particular time of the day? Had they emptied their bowel and bladder?

Line 136 if measurements were conducted by healthcare practitioners and not participants, comment on the detection bias – were the assessors blinded to which group each participant belonged to

Results

Line 154 address attrition bias/completeness of outcome data – Did you use intention to treat? What did you do about missing values? – Multiple imputation? Last carry forward? Did you show no differences between control and treatment group in terms of retention/attrition? Were the participants that dropped out different to the ones staying?

Line 157 “…had some college education or more”

Line 160 include a sentence stating that the two groups were comparable in terms of demographics

Line 162 for the ones that self-identified multiple races, how did you group them?

Line 163 move p to after “similar”, i.e. similar (p=0.580)

Line 171 were the >35yo any different to the 18-24yo and/or the 25-34yo in terms of demographics?

Results/general comment: Any report on compliance with intervention aside the attrition rate?

Discussion

Line 201 strengths: You may consider the attrition rate if you used an intention-to-treat analysis

Line 212 limitations: It may also be the opposite, with overestimation. See comments above in methods section and consider expanding

Line 214 in your methods you mention that 80 would be enough. After the drop out you have 72. What was the statistician’s advice? Did you analyse all using intention-to-treat?

Line 215 across more locations yes. But reconsider if a longer duration would facilitate more enrolment. You only enrol at baseline, thus timeline does not affect this as it is a single timepoint. Unless if you postulate that the longer duration would entice more participants. I would argue the opposite. Also the possibility of drop-out increases with length of time. Reconsider

Line 219 yes, agreed. Note though that your demographic i.e. low-income with higher obesity prevalence is also the one with higher short message service usage – you may want to consider commenting on this

Line 230 great comment

Lines 258-260 this is a result. It should first be reported in the results section. Include a sentence about participants’ feedback in the results section before discussing it here as it is currently not supported by data in the results section

Lines 260-263 ok yes this has been proposed before but other studies have also shown that decreased frequency associated with sms personalisation and tailoring and bidirectionality was associated with improved outcomes. But yes it has been discussed that the more frequent the reporting of weight via text messaging had been, the more weight was lost at 12 months as a percentage of initial weight - Consider discussing both

General comments:

  1. In addition to text message frequency, dose, duration of intervention, personalisation and bidirectionality you may want to consider discussing level of interactivity, dialogue mode initiation (researcher vs participant), inclusion of additional components (e.g. other studies have employed SMS in conjunction with a Facebook support group), prompts (e.g. the Loozit study from Australia identified that healthy eating messages and those concluding with ‘please reply’ elicited the highest reply rates from adolescent participants), tone of the language (e.g. a US study found that obese adolescents preferred positive, encouraging and direct messages).
  2. Finally, especially since you are assessing a low-income population, a cost effectiveness analysis should be recommended and future studies should aim to report the costs of such programmes in addition to the clinical outcomes.

Reviewer 2 Report

The study by Holmes et al. used an RCT design to investigate the effect of short message service (SMS) intervention (on nutrition and physical activity) on gestational weight gain (GWG) among a low-income population in Hawaii. The study found that the SMS intervention does not have a statistically significant effect on GWG absolute value or the percentage of excessive GWG based on the IOM recommendations. The main strength of the study is the randomized controlled design. Some limitations of the study include self-reported pre-pregnancy weight, a small sample size, and the lack of data on dietary intake and physical activity. The statistical approach is standard, and the manuscript is well-written.

  1. Line 17: please spell out the full name of WIC.
  2. Line 39: “with GWG reduced in one group or not at all.” I would suggest clarifying what “one group” refers to here. Does it mean the intervention group in previous intervention studies?
  3. Line 45: The statement regarding low-income groups more likely to have excessive GWG applies to the U.S. and other high-income settings. However, in low- and middle-income countries (LMICs), the opposite situation is true, and much of the focus in LMICs is to prevent inadequate GWG. I would suggest clarifying the focus and generalizability of this manuscript to the low-income population in high-income countries.
  4. Line 80: All of the participants were overweight or obese (BMI: 25-45) in the first trimester. I would appreciate some clarification on why this is one of the inclusion criteria. Is this because the focus was on preventing obsessive GWG among overweight and obese women (as evidenced by the Discussion section)? This is an important characteristic of the study that has implications regarding the generalizability. I would suggest making this point apparent throughout the manuscript, because "preventing excessive GWG" does not automatically imply that the target population will exclusively be overweight/obese women.
  5. Related to the previous point, what was the range of gestational age for "in the first trimester." The participants were enrolled 15-20 weeks of gestation, so I would presume that the first-trimester weight or BMI was also self-reported similar to pre-pregnancy weight?
  6. Lines 100-103: A conceptual framework or theory-of-change graph will be great to demonstrate the components of the intervention, pathway, intermediate outcomes, and the ultimate outcome of GWG.
  7. Lines 103-107: The messages on physical activity were created based on the ACOG PA recommendation, which is very clear with specific recommendations on duration and intensity. However, the IOM 2009 Guidelines do not explicitly make specific recommendations regarding caloric intake or dietary intake. I would appreciate some additional clarification on how the IOM 2009 Guidelines informed the design of nutrition messages. This also needs to be clarified in the Abstract (lines 18-20), which again implies that the nutrition messages were based on IOM 2009 Guidelines, in a similar way as the physical activity messages were based on the ACOG recommendations.
  8. Line 135: What was the range of gestational age when the last weight was taken? Was it immediately prior to the onset of delivery? How much variability is there regarding the timing of last weight measurements? Additional information on how the weight was measured and by whom will be helpful.
  9. Table 1 includes the IOM recommendations for women who were normal weight before pregnancy. This column seems redundant and contradicts the inclusion criterion of first-trimester BMI 25-45.
  10. Lines 147-149: The list of covariates eventually included in the final multivariate model should be included. It is also important to mention that, since the intervention was randomly assigned, the covariates in the multivariate model were included as precision variables to include statistical precision instead of confounders for the intervention effect.
  11. Lines 163-177: I suggest organizing these two paragraphs into tables as well for the ease of reading.
  12. Lines 184-185: More than half of the women exceeded IOM recommendation in the intervention and control groups, so I would not use the word "only" here. Also, it is not clear what point this sentence is trying to get across.
  13. The study population was labeled as a low-income population. However, no information is provided on the income or SES level of the participants (besides education level in Table 1), and no inclusion or exclusion criteria pertain to this. More context and background information about the SES of the study participants or the underlying base population will be helpful.
  14. The manuscript mentions the advantages and increased feasibility of technology-based interventions versus traditional, face-to-face interventions. Therefore, I think the study missed the opportunity to actually making the comparison in this study by including a third arm of face-to-face interventions.
  15. Lines 205-206: Please clarify how this is considered a strength of the study as all women should be receiving the standard of care regardless.

Round 2

Reviewer 2 Report

The authors have adequately addressed all of my questions and suggestions. I do not have additional feedback for this manuscript.